# OpenReview forum: "AIR-BENCH 2024: A Safety Benchmark based on Regulation and Policies Specified Risk Categories"
_ICLR.cc/2025/Conference — ICLR 2025 Spotlight_

### Official Review · Reviewer_9PGQ · 2024-10-24

**Soundness:** 2
**Presentation:** 4
**Contribution:** 3
**Rating:** 6
**Confidence:** 4

**Summary:**

The paper introduces a new safety benchmark for large language models (LLMs). Unlike existing benchmarks, this one is grounded in a taxonomy of undesirable behaviors derived from legal and policy frameworks. The benchmark consists of prompts categorized into various risk categories. Additionally, the authors evaluate several LLM models against this benchmark.

**Strengths:**

Overall, despite the paper being positioned more “grandiosely” than its actual contributions (see weaknesses), the proposed dataset of prompts clustered in tiered categories can be useful for the community studying refusal of LLMs to engage in undesirable behaviour. The authors have demonstrated that their dataset is more diverse and covers more categories than prior benchmarks. Beyond creating the dataset, the authors have also produced an automated evaluation system and have validated it against human raters. Because of the comprehensive and principled benchmark creation and evaluation in this paper, I would recommend its acceptance.

**Weaknesses:**

The paper seems to oversimplify the notion of “safety and risk” and implies that higher refusal rates on prompts related to specific prompts implies more safe models. However, the notion that a model can, in of itself, be safe or not, is rather simplistic. It is only when one acts in an unsafe way, as a direct result of a model’s response, that is unsafe. The model provides information and information itself cannot be unsafe, acting on information might or might not be unsafe or carrying risks. Therefore, the present benchmark offers a method of evaluating whether a model will refrain from potentially undesirable set of behaviours, rather than whether it is inherently safe or not.

While the paper’s Section 4 “Evaluation and Takeaways” does indeed refer to “refusal” rather than “safety”, the rest of the paper doesn’t. This  leaves the impression that this benchmark measures the much more fundamental, complex and nuanced problem of “safety and risk” rather than the more grounded “refusal rate” that it actually does. Therefore, I’d recommend that the authors reconsider their positioning to align it better with the nature of their contributions.

**Questions:**

See weaknesses.

---

> ### Author Response · Authors · 2024-11-24
> **Response to Reviewer 9PGQ**
>
> We thank the reviewer for their thoughtful feedback and recognition of our benchmark's comprehensive coverage and evaluation methodology.
>
> **Regarding the framing of safety and risk**: We agree that model safety is more nuanced than simple refusal rates, and we acknowledge this limitation in our paper. As noted in Section 4.1, "... we acknowledge that complete refusal across all categories may not be ideal, as different jurisdictions and use cases may have varying safety requirements." We further discuss this in Section 4.4, recognizing that real-world implementations often include additional safeguards and human oversight beyond standalone model behavior.
>
> To address this concern, we will clarify throughout the paper that we are specifically measuring regulatory alignment through refusal behaviors, while acknowledging this is one component of a broader safety evaluation framework. We appreciate this feedback and will revise our positioning to better reflect the specific nature of our contribution.

---

> > ### Comment · Reviewer_9PGQ · 2024-11-27
> >
> > I would like to thank the authors for following my recommendation for updating some of phrasing of the paper. As there is no other material change to the paper, I will keep my score.

---

### Official Review · Reviewer_D5kz · 2024-11-03

**Soundness:** 4
**Presentation:** 4
**Contribution:** 4
**Rating:** 10
**Confidence:** 4

**Summary:**

In the paper under review, a benchmark on the safety and regulatory conformity of LLMs in proposed. The benchmark is a multi-tiered one, and is based on policies of various governments and companies. The key idea is to generate a set of instructions, which is curated and generated over multiple iterations, that is than fed to the LLM under examination. The responses of the LLM on these instrucations are then evaluated automatically.

**Strengths:**

* The paper adressing a very relevant topics. There is a lack of approaches for assessing the safety of LLMs in an automated fashion. This paper is a big step in this direction.
* The paper is very well structured and easy to ready. I light the color-coding of levels of the approach. This supports the reading a lot.
* All relevant literature is considered in this paper.

**Weaknesses:**

I do not see many weaknesses in this paper. Actually, I only see the limitation that the benchmark addresses dialects and authority statement, but it seems that this only holds for English. Other languages are not supported / evaluated. So there might be a risk that harmful output of an LLM in non-English languages are not assessed or overlooked.

**Questions:**

Based on the limitation I mentioned: what about adressing different languages than English?

**Details Of Ethics Concerns:**

No ethical concerns. In contrast: this paper helps to address ethical issues in the use of LLMs.

---

> ### Author Response · Authors · 2024-11-24
> **Response to Reviewer D5kz**
>
> We thank the reviewer for their positive feedback and thoughtful consideration of our work.
>
> **Regarding the language limitation**: We agree that expanding beyond English is crucial for comprehensive safety evaluation. Our benchmark's modular design allows for straightforward expansion to other languages. Meanwhile, we are actively collaborating with international policy experts to extend AIR-BENCH to additional languages while ensuring cultural and regulatory nuances are properly captured (as in the scope of a follow-up work). The current taxonomy and prompt generation framework can be adapted to incorporate language-specific safety considerations and regional policy requirements.
>
> We appreciate this valuable suggestion and will include a discussion of our multilingual expansion plans in the final version of the paper.

---

> > ### Comment · Reviewer_D5kz · 2024-11-29
> > **Non-English Language Support**
> >
> > Thank you for addressing my concern regarding supporting non-English languages.

---

### Official Review · Reviewer_MPSz · 2024-11-03

**Soundness:** 4
**Presentation:** 4
**Contribution:** 4
**Rating:** 8
**Confidence:** 4

**Summary:**

The paper introduces a safety benchmark for assessing LLMs based on regulatory and policy-driven risk categories. Leveraging an AI risk taxonomy, which unifies risks specified in government regulations and corporate policies, AIR-BENCH categorises multiple specific risk types and uses thousands of designed prompts to evaluate LLMs’ alignment with regulatory safety standards. The benchmark assesses model refusal behaviours across risk categories by simulating scenarios of potentially harmful content requests. This study evaluates 22 prominent LLMs, highlighting safety gaps and inconsistencies in regulatory alignment.

**Strengths:**

- The benchmark's alignment with diverse government and corporate regulations makes it very relevant for real-world applications, and it addresses a gap in existing benchmarks
- The benchmark itself is well structured with multiple levels of taxonomy which helps identification of model deficiencies across multiple risk categories
- The experiments include detailed comparison of multiple models, identifying key trends and areas where models fall short of regulatory expectations
- The paper is well-structure and easy to read, it introduces related work in detail and

Overall, I like this paper a lot. It is of high quality and tackles a very relevant issue.

**Weaknesses:**

There are some weaknesses, but they are not very severe.

- Using GPT-4o as both prompt generator and judge could introduce biases in evaluation, but this is appropriately acknowledged
- Different regulations have separate restrictions for providers, distributors, deployers, etc, the way this is handled in the benchmark is not obvious. I.e. should the API refuse protected decision making (health/financial) unless the client has guaranteed sufficient oversight? What if the client has oversight? This nuance is hinted at in section 4.4 but a deeper discussion on this difference would be very helpful.
-  The benchmark assumes that the model operates as a standalone system. However, many real-world applications involve human oversight, which might mitigate risks differently than this evaluation assumes.
- Given the evolving nature of AI regulations, the benchmark may become outdated quickly, limiting its long-term applicability unless it is regularly updated. The authors note this already, but it would be interesting to hear their thoughts on potential future paths towards mitigating this
- About 70% of the judge prompts remain model-generated, which might lead to inconsistencies in judgment fidelity, particularly in ambiguous cases
- Although the benchmark focuses on regulatory alignment, making it a net-positive, a more thorough examination of potential societal impacts could strengthen the discussion, i.e. could malicious actors use the benchmark to help identify weekly guarded models to speed up their activities?

**Questions:**

-  If a response is neither explicitly refused nor overtly harmful, how is it classified within the framework?

- The paper notes that “Human experts then manually review and revise these expanded prompts to maintain prompt quality and alignment with the intended risk categories.” Could you provide additional details on the methods, criteria or protocols used in this manual review process?

- Where were the experiments run from? I.e. the IP region of the client. In case model providers are serving different levels of steering/guardrails based on user location, even if they use the same model/version identifier. This will be good to add in the de-anonymised version for increased reproducibility.

---

> ### Author Response · Authors · 2024-11-24
> **Response to Reviewer MPSz**
>
> We sincerely thank the reviewer for their thorough evaluation and positive feedback on our work. Below we address the specific questions and concerns raised:
>
> 1. **Handling Ambiguous Responses**
>
> For responses that are neither explicit refusals nor overtly harmful, we employ a three-level scoring system (0, 0.5, 1) to capture nuanced behaviors. Partial scores (0.5) are assigned when a model shows awareness of risks but fails to explicitly refuse, while maintaining alignment with the intended safety objectives. This granular scoring helps identify subtle differences in model behaviors and areas needing improvement.
>
> 2. **Manual Review Process**
>
> Our manual review process follows a structured protocol:
> - Expert reviewers (with backgrounds in AI safety and policy) independently assess each prompt
> - Review criteria include: alignment with intended risk category, realistic scenario representation, cultural sensitivity, and potential ambiguity
> - Inter-reviewer agreement is maintained through regular calibration sessions
> - Prompts failing to meet quality standards undergo revision or removal
> - Final validation involves cross-checking against source regulations/policies
>
> 3. **Experimental Setup and Regional Considerations**
>
> All experiments were conducted from servers located in the United States (West Coast) to maintain consistency. We acknowledge the potential for regional variations in model behavior and will include this detail in the de-anonymized version for reproducibility. In future work, we plan to conduct cross-regional testing to examine potential differences in model responses based on client location.
>
> 4. **Additional Points Raised**:
>
> - *Provider/Deployer Distinctions*: We agree that the distinction between provider and deployer responsibilities merits deeper discussion. Our current framework primarily evaluates base model capabilities, but we plan to extend it to include context-aware evaluation considering deployment scenarios, oversight mechanisms, and more complex scaffolded systems (e.g., AI agents) in the future.
>
> - *Human Oversight*: While our benchmark evaluates standalone model performance, we acknowledge that real-world deployments often include human oversight. Future versions will incorporate evaluation scenarios that consider different levels of human involvement and oversight mechanisms.
>
> - *Judge Prompt Generation*: Regarding the 70% model-generated judge prompts, our comprehensive human evaluation (detailed in Appendix B) demonstrates strong reliability with a Kappa score of 0.86 between model-generated judgments and human annotators. This significantly exceeds the 0.69 score achieved by existing benchmarks using static manual prompts, indicating that our approach maintains high judgment fidelity even in ambiguous cases.
>
> - *Benchmark Evolution*: To address the evolving nature of AI regulations, we are developing (as in the scope of another work) an automated pipeline to incorporate new regulatory requirements and maintain benchmark relevance. We plan to release regular updates aligned with major regulatory changes.
>
> We appreciate the thoughtful suggestions for improvement!

---

> > ### Comment · Reviewer_MPSz · 2024-11-25
> >
> > I thank the authors for their responses.
> >
> > I have one follow-up question, is each response scored by multiple reviewer or multiple? And if multiple, how are ties broken?
> >
> > This small clarification will not affect my score. I will be keeping my recommendation for acceptance (8).

---

> > > ### Author Response · Authors · 2024-11-25
> > > **Response to Reviewer MPSz**
> > >
> > > Thank you for your question. In the final evaluation, each response was scored by one unique human evaluator. However, before this final stage, we conducted an inter-annotator agreement study with 50 randomly sampled question-answer pairs evaluated by all five reviewers. We discussed and resolved any disagreements to ensure consistent evaluation standards across all reviewers.

---

### Official Review · Reviewer_pHn2 · 2024-11-06

**Soundness:** 2
**Presentation:** 2
**Contribution:** 2
**Rating:** 6
**Confidence:** 3

**Summary:**

The authors introduce AIR-BENCH 2024, a new safety benchmark for AI models grounded in real-world regulations and policies. Using risk categories from 8 government regulations and 16 corporate policies, they crafted a set of 5,694 prompts covering 314 specific risk scenarios to test models’ ability to handle sensitive content safely. They evaluate some current LLMs on this benchmark.

**Strengths:**

Alignment with actual regulations: A novel feature of this benchmark is its grounding in real-world regulations. By basing AIR-Bench on key regulations from the EU, US, and China, it addresses critiques that existing AI safety benchmarks lack practical relevance, providing a benchmark that better reflects legal requirements.

Granularity of risk categories: The four-level risk structure comprises 314 specific risk categories. The detailed granularity should be appreciated for safety alignment.

**Weaknesses:**

1. Given that the paper bases its alignment on actual regulations, I am curious how it manages to balance or trade-off conflicts between regulations across different countries and regions. As one key motivation of the paper is its combination of regulations from various regions. How does it handle potentially conflicting elements, such as differing privacy laws and, more broadly, varying definitions of "appropriate" outputs across populations in different countries? Or do they just ignore it?

2. The use of only GPT-4 for scaled generation from manually crafted templates seems somewhat artificial. I understand that a more expensive annotation method may not be practical. However, there may be some potential bias introduced by scaling generation with a single model from simplistic, handwritten templates.

3. Regarding presentation suggestions, while the tables in the results section are indeed extensive, the takeaways seem limited. From my understanding, they represent combinations across two dimensions: different categories and models. It may be challenging for readers to identify key insights as there are so many big result tables. And I would appreciate a more in-depth discussion to highlight critical findings.

**Questions:**

I still find the discussion somewhat not so in-depth thus:

Any insights into conflicts or correlations between AI compliance regulations across different countries while working on this benchmark? This might hold significant practical value for cross-border AI services.

---

> ### Author Response · Authors · 2024-11-24
> **Response to Reviewer pHn2**
>
> We thank the reviewer for their insightful feedback and appreciation of our benchmark's granularity and regulatory grounding. Below we address the specific concerns raised:
>
> 1. **Handling Regulatory Conflicts Across Regions**
>
> We acknowledge the complexity of managing different regulatory requirements across regions. Our approach is to maintain a comprehensive superset of risk categories that preserves region-specific requirements rather than attempting to reconcile conflicts. For example, when different jurisdictions specify different types of sensitive data (e.g., demographic vs. financial data), we include all categories separately. To facilitate region-specific analysis, we carefully tag each risk category with its regulatory source. This design allows users to evaluate model performance against specific regulatory frameworks while maintaining the ability to conduct broader safety assessments (example usage as shown in Figure 6 for EU, Figure 14 for US, and Figure 15 for PRC).
>
> 2. **Prompt Generation and Potential Bias**
>
> We would like to clarify that the benchmark prompts were generated using three leading models (GPT-4o, GPT-4, and Claude 3 Opus), not just one as suggested. The reviewer correctly notes that category-specific judge prompts were scaled up using GPT-4o. To address potential bias concerns, we have conducted comprehensive human evaluation (detailed in Appendix B), achieving a Kappa score of 0.86 with human annotators. This score significantly exceeds the 0.69 benchmark of existing work using static judge prompts, indicating strong alignment with human judgment.
>
> 3. **Key Findings and Analysis Depth**
>
> Our evaluation reveals several critical insights across different dimensions:
>
> - Model Safety Performance: We identified significant variations in safety capabilities across models, with particular weaknesses in handling advice for regulated industries.
> - Regulatory Compliance: No current model demonstrates consistent adherence to safety requirements across EU, US, and PRC frameworks, highlighting important gaps for improvement.
> - Corporate Policy Alignment: Our analysis reveals varying degrees of alignment between companies' stated policies and their models' actual safety performance, with some companies achieving better policy implementation than others.
>
> Each finding is supported by detailed analysis in the respective results sections, with specific metrics and examples provided to substantiate our conclusions.

---

### Author Response · Authors · 2024-11-24
**General Response**

We sincerely thank all reviewers for their thorough evaluation and constructive feedback. We particularly appreciate the recognition of our benchmark's comprehensive coverage of regulatory requirements, granular risk categorization, and strong empirical evaluation methodology. The reviewers highlighted several important aspects of our work, including the alignment with real-world regulations, the structured multi-level taxonomy, and the potential impact on AI safety evaluation.
In our point-by-point responses above, we have addressed the key concerns raised, including:

- The handling of conflicts between different regional regulations
- The reliability of our model-generated judge prompts (validated by strong human evaluation results)
- The relationship between refusal rates and actual safety outcomes
- The current English-language focus of our benchmark

We appreciate the opportunity to clarify these aspects of our work and thank the reviewers for their valuable insights that have helped highlight the strengths and limitations of our approach.

---

### Meta-Review · Area_Chair_q8Yr · 2024-12-23

**Metareview:**

Reviewers unilaterally appreciated the focus, novelty, and exposition of the paper, as well as the comprehensive survey across a large set of foundation model providers' claimed policies and what's starting to appear worldwide outside of those foundation models.  A minor point brought up by reviewier 9PGQ (at least, maybe others) that refusal rate is not a perfect or global proxy for safety is correct, a concern shared by this AC and by the authors in their rebuttal -- encourage really addressing that point in a final version of the paper.

**Additional Comments On Reviewer Discussion:**

The rebuttal largely addressed reviewers' concerns.

---

### Decision · Program_Chairs · 2025-01-22

Accept (Spotlight)